# Withdrawal Performance of Nails and Screws in Cross-Laminated Timber (CLT) Made of Poplar (*Populus alba*) and Fir (*Abies alba*)

**DOI:** 10.3390/polym14153129

**Published:** 2022-07-31

**Authors:** Farshid Abdoli, Maria Rashidi, Akbar Rostampour-Haftkhani, Mohammad Layeghi, Ghanbar Ebrahimi

**Affiliations:** 1Department of Wood and Paper Science, Faculty of Natural Resources, University of Tarbiat Modares, Tehran 14117-13116, Iran; 2Centre for Infrastructure Engineering, Western Sydney University, Sydney 2000, Australia; 3Wood Science and Technology, Department of Natural Resources, Faculty of Agriculture and Natural Resources, University of Mohaghegh Ardabili, Ardabil 56199-11367, Iran; 4Department of Wood and Paper Science and Technology, College of Natural Resources, University of Tehran, Karaj 14179-35840, Iran; mlayeghi@ut.ac.ir (M.L.); gh.ebrahimi@ut.ac.ir (G.E.)

**Keywords:** cross-laminated timber, withdrawal resistance, nails, screws, loading direction, layer arrangement, bridges

## Abstract

Cross-laminated timber (CLT) can be used as an element in various parts of timber structures, such as bridges. Fast-growing hardwood species, like poplar, are useful in regions where there is a lack of wood resources. In this study, the withdrawal resistance of nine types of conventional fasteners (stainless-steel nails, concrete nails and screws, drywall screws, three types of partially and fully threaded wood screws, and two types of lag screws), with three loading directions (parallel to the grain, perpendicular to the surface, and tangential), and two layer arrangements (0-90-0° and 0-45-0°) in 3-ply CLTs made of poplar as a fast-growing species and fir as a common species in manufacturing of CLT was investigated. Lag screws (10 mm) displayed the highest withdrawal resistance (145.77 N), whereas steel nails had the lowest (13.13 N), according to the main effect analysis. Furthermore, fasteners loaded perpendicular to the grain (perpendicular to the surface and tangential) had higher withdrawal resistance than those loaded parallel to the grain (edge). In terms of the layer arrangement, fasteners in CLTs manufactured from poplar wood (0-45-0°) had the greatest withdrawal resistance, followed by CLTs manufactured from poplar wood in the (0-90-0°) arrangement, and finally, those made from fir wood in the (0-90-0°) arrangement. The fastener type had the most significant impact on the withdrawal resistance, so changing the fastener type from nails to screws increased it by about 5–11 times, which is consistent with other studies. The results showed that poplar, a fast-growth species, is a proper wood for manufacturing CLTs in terms of fastener withdrawal performance.

## 1. Introduction

Sustainable building approaches using renewable resources such as wood and wood-based products have grown in popularity in recent decades [1,2,3]. Cross-laminated timber (CLT) is an engineered wood product (EWP) formed from sized lumbers and orthogonally laminated. This relatively new form of EWP is reported to be extensively employed in many projects, including mid-rise and even high-rise buildings, due to its unique cross-wise layups that can tolerate large loads and stresses either in-plane or out-of-plane [4,5,6]. CLT might be regarded as a stand-alone element because of its high degree of prefabrication and load-carrying capabilities. It is commonly utilized as a wall, floor diaphragm, roof, and other structural components. Various connectors, such as angle brackets and hold-downs with fasteners (nails and screws), might be used to join these parts. Connections, like a fuse in an electrical circuit, are frequently the source of ductility and energy dissipation in the structure in case of overloading because of the brittle failure behavior of wood when stressed in tension or shear [7]. The connections between structural parts must reinforce the structures to withstand gravity or vertical loads induced by self-weight, live loads, wind, and seismic loads, and then transmit these forces to the foundation [8,9,10].

Most failures begin at the connections, usually the weakest sections of timber structures [11]. Hence, it is vital to investigate how connections and fasteners behave under different loads. Nails or screws in CLT connectors are exposed to lateral and withdrawal loads, or a combination of both. When exposed to withdrawal loads, fasteners may pull out (withdraw) of wooden members or rupture in tension. Head pull-through is uncommon in CLT connectors.

The diameter, thread geometry, penetration depth of a fastener, and load-to-grain angle are the main factors that influence the withdrawal resistance of a fastener [12,13,14]. The latter is owed to the orthotropic nature of wood and wood-based products like CLT. Commonly, withdrawal loading is characterized as parallel or perpendicular to the grain, where the latter includes both radial and tangential loading, as shown in Figure 1.

In CLT, the loading directions are characterized as edge loading and surface loading, where edge loading can occur parallel to the grain (L) or in a tangential direction (T). The type of loading is mainly governed by the application of the CLT panel (wall-to-wall, wall-to-floor, etc.).

Uibel and Blaß [15,16,17] first studied the lateral and withdrawal resistance of screws, nails, and dowels in CLT made of European spruce (*Picea abies*). Since these first studies in the early 2000s, several more studies have examined the impacts of different parameters on the withdrawal performance of fasteners in CLT. Design guidance for CLT connections is given in several CLT Handbooks [18,19,20] and by CLT and fastener manufacturers. Yet, due to the many possible parameter combinations, most studies only considered a limited combination of grain orientations; loading directions; fastener types and fastener features (including thread geometry, diameter, and penetration depth); and wood species. The selected literature related to the present study is summarized here to contextualize the experimental program presented in this paper. Li et al. [21] investigated the withdrawal resistance of bamboo scrimber specimens modified by embedment length, screw diameter, and screw angle. They revealed that unlike wood, the tensile strength and stiffness of the bamboo scrimber in the radial and tangential directions are comparable. By increasing the density of CLT, the withdrawal capacity and strength of self-tapping screws enhanced from 19.7 to 57.3% [22].

Several studies examined the influence of the load-to-grain angle on the withdrawal capacity in sawn timber. One of the most comprehensive works on fully threaded self-tapping screws was published by Blaß and Bejtka [23] considering embedment, penetration depth, load-to-grain angle, lateral loading, and minimum fastener spacing requirements, all in sawn spruce (*Picea abies*).

Teng et al. [24] investigated the effect of the load-to-grain angle on the withdrawal resistance of screws and nails in sawn larch (*Larix gemlimii Rupr.*) and spruce (*Picea glauca*). They concluded that the insertion angle had a significant effect; however, there was no significant difference between radial and tangential directions. Furthermore, they found a positive correlation between density and withdrawal strength. Ringhofer et al. [25] studied the effect of combined axial and lateral loading in sawn Norway spruce (*Picea Abies*), highlighting the influence of the penetration depth and potential splitting for screws inserted parallel to grain (end grain).

Similar observations were made for CLT, where the withdrawal strength of self-tapping screws in the CLT side face (surface) is much greater than that of screws in the CLT narrow face (edge) [14]. This is also reflected by the failure modes observed in these different loading directions. Li et al. [5] investigated the withdrawal resistance of a single self-tapping screw in CLT manufactured from radiata pine (*Pinus radiata*). According to their results, column-like pull out of timber components was observed when self-tapping screws were axially withdrawn parallel to the grain direction, whilst the tearing failure of adjacent fibers occurred when they were withdrawn perpendicular to the grain direction.

However, the loading situation in CLT is often more complex since fasteners may penetrate one or more layers leading to more complex stress states, especially when fasteners are loaded in a combination of axial withdrawal and lateral shear loading. Hossain et al. [26] studied combined axial and lateral loading for screws in spruce-pine-fir CLT. Brown et al. [27] studied the effect of different angles and penetration lengths of inclined screws in CLT made of New Zealand radiata pine (*Pinus radiata*) and Douglas fir (*Pseudotsuga menziesi*), giving recommendations on the maximum penetration length to avoid screw rupture, and the ratio of different load-to-grain angles to optimize ductility [28].

Yet, even more, simple parameters, such as timber density, require more careful consideration in CLT. It is generally accepted that there is a positive correlation between fastener withdrawal strength and timber density [29]. However, in CLT, different layers may exhibit different densities. Mahdavifar et al. [12] discovered that the density of the face layer had a substantial impact on the fastener withdrawal resistance of wood screws and ring shank nails in hybrid CLTs produced from Douglas fir and lodgepole pine.

In CLTs made of Japanese larch (Larix kaempferi (Lamb.) Carr.), the withdrawal strength of self-tapping screws increased with increasing the timber density and effective length, and the withdrawal failure mode was a mix of shear cracks parallel to the grain due to fiber bending and tension perpendicular to the grain [30].

Furthermore, the surface conditions of the fastener have a significant impact on withdrawal capacity. Izzi et al. observed that a threaded shank enhanced the withdrawal capability of nails in CLTs manufactured from spruce when compared to smooth shank nails [31]. The friction between the threaded shank and the surrounding wood in CLT then determines the load-bearing mechanism of the annular-ringed shank nailing joints loaded in withdrawal [32]. According to Ceylan and Girginannular [33], threads make fasteners easy to insert and hard to pull-out. Rezvani et al. [34] showed that fully threaded screws in angle bracket connections outperformed partially threaded screws in terms of uplift, in-plane shear, and out-of-plane stress. D’Arenzo et al. found that adding inclined fully threaded screws in the bottom corner of traditional angle brackets significantly increased the performance for tensile loading, avoiding the usual low withdrawal resistance of the annular-ringed nails in the bottom plate [35].

In addition to fastener diameter and surface conditions, fastener penetration depth needs to be considered. Li et al. [36] found that increasing penetration depth by 50 mm (75 mm vs. 125 mm) increased the withdrawal resistance of radiata pine CLTs in the narrow face with self-tapping screws. In wood-plastic composite panels, screw withdrawal resistance rose when screw diameter, loading rate, and penetration depth increased [37].

Finally, the wood species used for CLT manufacturing significantly impacts their properties. Some hardwood species, such as poplar, grow fast, which is advantageous in countries with a shortage of wood supplies, such as Iran [38]. The shortage of wood supplies is a global problem. Although several research studies have investigated the withdrawal performance of CLTs made of softwood species such as pine, fir, and spruce, there are few findings on CLTs made of poplar wood. Most studies so far focused on the structural properties of poplar CLT, including flexural behavior and rolling shear [39,40,41,42,43,44].

The material properties of wood and wood-based products, such as density, manufacturing technique, and environmental conditions, have been shown to affect the withdrawal behavior of fasteners [45]. Hübner et al. [46] confirmed that the withdrawal resistance is affected by differences in wood species (or material properties). Moreover, Taj et al. [47] reported that anatomical diversity in wood has a substantial influence on fastener withdrawal resistance. According to Ringhofer et al. [13,48], the density of hardwood boards, geometric qualities and screw penetration depth, the number of layers that the screw penetrated, the angle, and gap insertion are significant factors to consider in the withdrawal performance of CLTs. Furthermore, methods for determining the withdrawal strength of self-tapping screws in solid and EWP were proposed.

Apart from the aforementioned characteristics, Yermán et al. [49] evaluated the withdrawal resistance of nails in modified and unmodified pine wood in terms of cyclic moisture changes. The results indicated that under moisture fluctuations, nail withdrawal ability seems to be lost owing to a combination of corrosion and the mechano-sorptive process of the nails backing out with alternating wetting and drying. When self-tapping screws were inserted in CLTs made from spruce, the withdrawal resistance was reduced by increasing the moisture [50].

In summary, most studies are on the withdrawal resistance of self-tapping screws and nails in softwood CLTs made of pine, fir, spruce, or larch. No research has been conducted on connections in poplar CLT to the authors’ knowledge. As a result, the aim of this study is to examine the withdrawal performance of 3-ply CLTs made from poplar as a fast-growing species with various fasteners (seven types of screws and two types of nails) in three withdrawal loading directions (parallel to grain and perpendicular to the grain, both radial and tangential) in two-layer arrangements of 0-90-0° and 0-45-0°. Furthermore, the findings are compared to data obtained from CLTs made of fir, which is a common softwood species used in CLT manufacturing with a similar density to poplar.

## 2. Material and Methods

### 2.1. Wood and Manufacturing of CLT

Poplar *(Populus alba)* and fir *(Abies alba)* wood with oven-dried densities of 381 and 390 kg/m3, respectively, were used in this research. For poplar, the modulus of elasticity, modulus of rupture, and shear strength parallel to the grains were 7380 MPa, 59 MPa, and 4.96 MPa, respectively. Similarly, they were 6658 MPa, 59.6 MPa, and 6.52 MPa for fir, respectively.

The poplar logs were cut into plain sawn boards with dimensions of 2000 mm × 110 mm × 25 mm (Length × Width × Thickness). Plain sawn fir boards with the same dimension were also prepared. Afterward, boards were air-dried at a temperature of 20 °C and a relative humidity of 65% until a constant weight was achieved. After drying, the boards were sawn and planed to the final cross-section size of 90 mm × 16 mm (Width × Thickness).

Poplar and fir boards without any wood defects or significant knots were selected and layered in 0/90/0° and 0-45-0° arrangements for poplar CLTs, and 0-90-0° for fir CLTs. The boards were both surface and edge-glued with one-component polyurethane adhesive (at a spread rate of 300 g/m^2^) and cold-pressed for 150 min at a pressure of 1 MPa using hydraulic equipment. The CLT panels were stored for several weeks at a relative humidity of 65% and a temperature of 25 °C.

### 2.2. Fasteners

Seven types of screws and two types of nails as fasteners were employed in this investigation. The characteristics of fasteners are given in Table 1. Note that in addition to typical timber fasteners, concrete nails and screws, as well as drywall screws, were employed to study the effect of different surface conditions. The different withdrawal directions with respect to the CLT samples are displayed in Figure 2.

Smaller blocks with a dimension of 75 mm × 75 mm × 48 mm were cut from the CLT panels, and a fastener was inserted in the three directions as shown in Figure 2a: parallel to the grain in the middle layer (L), the tangential direction of the middle layer (T), and perpendicular to the surface of the CLT panel (S). Fasteners were inserted at a 32 mm penetration depth into the CLT samples, as depicted in Figure 2b. The fasteners were placed to eliminate any gaps and to meet the requirements of boundary conditions, and end and edge distances stipulated in previous studies [14,51]. In other words, it means that the distance between the screws in different directions was such that they did not affect each other.

### 2.3. Experimental Setup

In order to measure the withdrawal resistance of the fasteners, CLT samples were installed in an Instron testing machine model 4486 (Norwood, MA, USA), (Figure 3). A constant displacement rate of 6 mm/min was applied according to ASTM D 1761 [52]. Finally, the withdrawal resistance of the fasteners was calculated according to Equation (1):W = P_max_/L(1)
where W is the withdrawal resistance of the fastener (N/mm), P_max_ is the maximum load (N), and L is the penetration depth of the fastener in CLT sample.

### 2.4. Statistical Analysis

Based on a completely randomized full factorial design, data from the experiments were statistically analyzed to determine the main and interaction effects between the nine fasteners, three loading directions, and three types of CLT panels, including poplar CLTs with two-layer arrangements (0-90-0° and 0-45-0°), and fir CLTs with only one arrangement of layers (0-90-0°). The full factorial design allows studying each independent variable with respect to the response variable (in this case, withdrawal resistance W), as well as the interaction between variables. Each variable may take on different values, and combinations of these values are called “treatment”. In the present study, 81 treatments were analyzed, each with six repetitions, for a total of 486 tests (162 specimens for each direction). Duncan’s multiple range test was performed to show the statistical differences between the treatment means at a 95% confidence level. SPSS software version 25 was used to conduct the statistical data analysis.

## 3. Results and Discussion

Table 2 gives the mean values of the withdrawal resistance of fasteners parallel to the grain (L) and perpendicular to the grain (T and S) in two arrangements for poplar CLTs and one arrangement for fir CLT. Screws displayed 7.5 to 11 times higher withdrawal resistance than nails. Concrete nails displayed a higher withdrawal capacity than steel nails. Furthermore, wood screws had the lowest withdrawal resistance among screws, whereas lag screws (10 mm) had the greatest. Fasteners in poplar CLTs with a 0-45-0° arrangement had the maximum withdrawal resistance, whereas the lowest withdrawal resistance was seen in fir CLTs. In terms of loading direction, fasteners perpendicular to the grain directions (S and T) demonstrated a higher withdrawal resistance than those parallel to the grain (L) direction. The highest withdrawal resistance was observed for fasteners inserted in the S direction.

Table 3 shows the analysis of variance of the main and interaction effects of CLT types, fastener types, and loading directions on the withdrawal resistance of CLTs. The findings revealed that CLT types, fastener types, and loading directions all had a significant influence on the withdrawal performance of the CLTs. Moreover, the interaction effects of CLT types * * fastener type, CLT types * loading direction, fastener type * loading direction, and CLT types * type of fastener * loading direction on withdrawal performance were significant.

### 3.1. Effects of CLT Types, Fastener Types, and Loading Directions on Withdrawal Resistance

The main effects of the CLT types, types of fasteners, and loading directions on the withdrawal performance of CLTs are shown in Figure 4A–C. According to the results, there was a significant difference in the withdrawal performance between CLT types, despite poplar and fir CLTS having similar specific gravity. CLTs manufactured with poplar wood showed higher withdrawal resistance in both arrangements than those made from fir wood (Figure 4A).

According to Figure 4A, poplar CLTs with the 0-45-0° arrangement had the highest withdrawal resistance (99.2 N), while those made from fir wood had the lowest (82.8 N). In other words, the withdrawal resistance of the CLTs made of poplar in the arrangement of 0-45-0° was 8.6 percent more than those in the arrangement of 0-90-0°. It might be due to the more substantial involvement of fasteners with timber fibers in 0-45-0°. Furthermore, the withdrawal resistance of poplar CLTs in the 0-90-0° configuration was 10.2 percent higher than that of fir CLTs. According to Brandner et al. [53,54], the pull-out strength increased linearly when the load grain angle rose from 0 to 30. Screw pull-out strengths varied between 19 and 24 percent for load-grain angles between 0 and 90.

Duncan’s test showed that there was a significant difference among the means. Variations in recorded values of fastener withdrawal resistance for two species with close specific gravity may be related to the anatomical structure of these two species. The presence of parenchyma rays in hardwoods like poplar may explain this difference [47], and it is hypothesized that these rays resist the withdrawal of fasteners. In other words, parenchyma rays are transverse elements in the wood that cause more involvement with the fasteners when they are exposed to the withdrawal force.

Figure 4B depicts the main effect of the loading direction on the withdrawal resistance of the CLTs. The findings showed a significant difference in withdrawal resistance of CLTs in all loading directions. More particularly, the withdrawal resistance was greatest when the loading was in the S direction (101.7 N). Loading in the L direction, on the other hand, resulted in the lowest withdrawal resistance (82.8 N), and the difference was 22.8 percent. Duncan’s test revealed a significant difference between the means of withdrawal in three directions. More details are discussed in the failure mode section. Since wood and wood-based products are orthotropic materials, anatomical variations in each direction can lead to significantly different properties such as the withdrawal performance in directions [47]. Li et al. [36] investigated the withdrawal resistance of self-tapping screws inserted in CLTs’ narrow faces (T and L). Their results indicated that the direction of CLT (T and L) affects the withdrawal resistance of self-tapping screws.

The withdrawal resistance of different fasteners in CLTs is shown in Figure 4C. The most effective variable in the withdrawal performance of CLTs was the fastener type. According to the results, steel nails had the lowest withdrawal resistance (13.13 N), while lag screws (10 mm) had the maximum withdrawal resistance (145.77), implying an 11 times difference. This confirms previous findings regarding the surface condition and diameter of different fasteners.

Duncan’s test indicated that there was no statistically significant difference between self-tapping screws (d = 6.4 mm), concrete screws (d = 7.8 mm), and lag screws (d = 7.8 mm), despite small differences in outer screw diameter.

On the other hand, the difference between wood, drywall, and SPAX galvanized screws was significant, despite smaller differences in the outer diameter (ranging between 4.2 and 4.7 mm). This may be owed to the different surface properties, including the difference between inner and outer diameter, thread gauge, and fastener material.

Concrete nails outperformed steel nails in terms of withdrawal resistance, with a statistically significant difference (47.2%). This result is consistent with the findings of Izzi et al. [31]. Concrete nails feature a low-profile thread on their shanks, providing a higher withdrawal resistance than steel nails, which do not have any threads. Furthermore, the diameter of the concrete nail is greater than that of the steel nail.

Overall, increasing the diameter of the fasteners enhanced their withdrawal resistance. Lag screws (10 mm) had the highest withdrawal resistance. Previous research found similar findings [12,21,33]. According to Gehloff [55], the diameter factor might significantly affect the fasteners’ withdrawal capacity. The withdrawal capacity of glulam rose by 70%, according to Abukari et al. [56,57], when the diameter of the screws was raised from 6 mm to 12 mm (100%).

Lag screws have a wide screw head diameter, which increases their resistance against head pull-through [58]. Self-tapping screws vary from primarily laterally loaded screws, such as hexagon head coach screws, in that they are optimized for loading in the axial direction [48].

Despite having a similar diameter, wood screws proved to be less resistant to withdrawal force than drywall screws due to their tapers, cut shank, and threads. According to Hoelz et al. [59], in addition to diameter, additional geometric characteristics such as flank distance and thread height affect fastener pull-out resistance. In this aspect, drywall screws had a greater flank distance and thread height than wood screws, resulting in a stronger resistance under the withdrawal load for drywall screws.

### 3.2. Interaction Effects of CLT Types, Fastener Types, and Loading Directions

Figure 5A displays the interaction effect of CLT types and fastener types under withdrawal force in CLTs. Fasteners in CLTs manufactured of poplar wood with the arrangement of 0-45-0° exhibited a stronger withdrawal resistance than other arrangements. Fasteners in 0-90-0° fir CLTs had the lowest withdrawal resistance. In terms of interaction between fastener type and CLT type, lag screws (10 mm) in the 0-45-0° arrangement had the highest withdrawal resistance (173.4 N), whereas steel nails in the 0-90-0° arrangement of CLTs made of fir wood had the lowest (10.7 N).

In general, fasteners exhibited greater withdrawal resistance as their diameter increased, and this trend was most pronounced for 0-45-0° poplar CLTs and 0-90-0° fir CLTs.

Altering the arrangement from 0-90-0° to 0-45-0° in CLTs made of poplar wood, the withdrawal resistance of steel nails and concrete nails changed 27% and 16%. Moreover, the withdrawal resistance of wood screws, lag screws (8 mm), and lag screws (10 mm) changed about 14%, 17%, and 24%, respectively. Furthermore, no significant difference was observed for other fasteners.

In changing the CLT types from poplar to fir, no significant difference was observed in the withdrawal resistance of concrete nails, drywall screws, and lag screws (8 mm); however, other fasteners showed 11–15% more withdrawal resistance in CLTs made from poplar wood in the arrangement of 0-90-0°.

Figure 5B depicts the interaction effect of CLT types and loading directions. The findings indicated that fasteners loaded perpendicular to the grains (S and T) had greater withdrawal resistance than fasteners loaded parallel to the grains (L). More specifically, the lowest withdrawal resistance (72.1 N) was reported in fir CLTs when the withdrawal was parallel to the grain (L). However, the greatest withdrawal load was observed in poplar CLTs with an arrangement of 0-45-0° (107.6 N) when the withdrawal loading direction was perpendicular to the surface (S).

CLT samples perpendicular to surface loading (S) had the highest withdrawal resistance, followed by samples with tangential loading direction (T), while the final one was parallel to the grain (L). The angle of fibers in the middle layer of poplar CLTs with the 0-45-0°-layer arrangement in contact with the fasteners was the same (45°) under both loading directions (L and T); hence, there was no significant difference between the mentioned loading directions. Figure 6d shows the related failure mode (inclined shear). In the CLTs made of poplar wood (0-90-0° and 0-45-0°), the biggest difference was observed in the L direction since altering the arrangement of the middle layer to 45° resulted in an increase of the withdrawal resistance (20%). In addition, between poplar and fir CLTs (both with 0-90-0° arrangement), the biggest difference was obtained from the S direction, while the lowest one was obtained from the L direction. In the poplar CLTs (0-90-0°), the difference between the S direction and L and T directions was about 33% and 21%, respectively. For the arrangement of 0-45-0° (poplar CLTs), these differences were about 12% and 15%, respectively. However, in the fir CLTs, the difference was about 26% and 1%, which means in softwoods such as fir, the difference in withdrawal resistance between the directions of S and T is insignificant. Conversely, for hardwoods such as poplar, the significant difference between S and T directions may be related to the placement of the parenchyma rays in the radial direction of the wood and perpendicular to the annual rings.

Figure 5C depicts the interaction impact of fastener types and loading directions. As expected, screws exhibited a higher withdrawal resistance than nails. Concrete nails had a higher withdrawal resistance than steel nails, although the difference was statistically insignificant in all loading directions and CLT types. Wood screws had the lowest withdrawal resistance among screws, whereas lag screws (10 mm) had the greatest, and this difference was statistically significant. Furthermore, the withdrawal resistance of fasteners perpendicular to the grain direction (T and S) was more than that of fasteners parallel to the grain direction (L). Except for lag screws (8 and 10 mm), the highest withdrawal resistance of the fasteners was recorded in the S direction loading perpendicular to the surfaces of the CLTs. By changing the fastener type, the withdrawal resistance changed more in the L loading direction compared to the others (S and T). The smallest change in the withdrawal resistance of the fasteners occurred in the T direction.

### 3.3. Failure Modes

Typical failure modes resulting from the withdrawal force parallel to the grain (L) are shown in Figure 6a–d; they are the column-like pull-out of wood fibers. Fasteners in this direction showed the lowest withdrawal resistance. It is worth differentiating between nails that simply spread fibers and screws that cut fibers. Li et al. [36] stated that shear stress formed around the threads under the withdrawal load parallel to the grain in the local wood regions, and nearby timber fibers were easily pulled out after stress limits were surpassed. The same failure mechanism was also reported by Ringhofer et al. [60].

The withdrawal resistance of drywall screws parallel to the grain (L) was found to be greater than that of the wood screws despite their outer diameters being similar (4.2 and 4.7 mm, respectively). Compared to drywall screws, wood screws have tapered shanks and cut threads. It seemed that this thread geometry significantly influenced shear failure and grain failure mode patterns as well as resulting in withdrawal resistance. The failure modes of the drywall and wood screws parallel to the grain (L) are shown in Figure 6a,b: drywall screws withdrew more wood fibers since their flank distance and thread height were greater than those of the wood screws, which is in agreement with previous studies [13,59]. The shape of threads also affected the shear failure along with the grain’s mode pattern. Concrete screws threads, for instance, operate as saw teeth, as can be seen in Figure 6c.

Hoelz et al. [59] stated that the failure mechanisms in the thread contact depend on the orientation of the wood fiber to the screw thread. In the present study, fasteners in the layer arrangement of 0-45-0° produced inclined shear failure along with the grain’s mode pattern (Figure 6d). In this case, the wood fibers emerged from one side of the fasteners due to the 45° angle of the wood fibers. None of the specimens exhibited fastener failure, e.g., head shear off or tensile rupture of the shank. In other words, all test failures included the withdrawal failures of the fasteners on wooden parts rather than the tensile failures of the fasteners themselves.

Typical failure modes resulting from the withdrawal force in the tangential (T) direction are shown in Figure 7a–d; they involve the tearing of adjacent fibers around the fasteners. All specimens in this direction showed the splitting failure of the wood layer’s fibers, and the failure only happened around the fasteners. Pang et al. [61] reported the same results. Fasteners in this loading direction (T) demonstrated a higher withdrawal resistance than in the L direction. According to Figure 7, greater damage occurred in CLT specimens in this direction than parallel to the grain (Figure 6). This is also reflected in the withdrawal resistance: the higher the withdrawal resistance of the fasteners, the more damage that occurred in the timber. For example, drywall screws exhibited a greater withdrawal resistance than wood screws in the T direction, and lifted more surrounding wood fibers than wood screws (Figure 7b–d). According to Li et al. [36], the difference in failure modes might be explained by the various stress levels at the interface between the screw threads and the wood components, leading to a combination of shear and tensile fiber failure when the fasteners were withdrawn perpendicular to the grain.

Typical withdrawal failure types caused by the force perpendicular to the CLT surface (S) are illustrated in Figure 8a–f, representing withdrawal failure accompanied by substantial fiber deformation. In other words, in this direction (S), the areas of the failures (Figure 7a–d) were larger than those in the L direction (Figure 6a–d). Fasteners showed the greatest withdrawal resistance in this loading direction (S). The existence of cross-section parenchyma rays on the tangential surfaces of CLT samples, particularly in poplar wood, may explain this [47]. As a result, more interactions occurred between wooden tissue and fasteners, resulting in an increased withdrawal resistance in the fasteners. As seen in Figure 8, greater damages occurred in the CLT specimens in this direction than in the other orientations (Figure 6 and Figure 7), especially for fasteners with high diameters. It means that fasteners with higher diameters cause more damage to CLT under the withdrawal load than fasteners with low diameters. According to Figure 8a,b, lag screws (10 mm) caused more damage than other fasteners, such as wood screws (Figure 8c), lag screws with an 8 mm diameter (Figure 8d,e), and drywall screws (Figure 8f) under the withdrawal load. Following that, lag screws with an 8 mm diameter (Figure 8d,e) caused more damage than wood screws (Figure 8c) and drywall screws (Figure 8f). It is worth noting that the position of fasteners in CLT members is important since they will be subjected to varied loads such as withdrawal. Consequently, when fasteners are positioned on the tangential surface, greater space between them is preferable, resulting in more resistance during withdrawal loading.

## 4. Conclusions

The following are the key conclusions that can be derived from the experimental results:-In terms of the main effect, fastener type was the most effective factor in the withdrawal performance of CLT. Following that were loading direction and CLT type, respectively.-In terms of the interaction effect, CLT type * fastener type was the most effective factor in the withdrawal performance of CLT. Following that were fastener type * loading direction and CLT type * loading direction.-In addition to fir wood, fasteners inserted in poplar (a fast-growing species) showed satisfactory withdrawal resistance.-Diameter had a great influence on the withdrawal resistance of the fasteners.-In terms of the loading direction, which is crucial in CLT connection design (wall to wall, wall to floor, etc.), fasteners in the S direction had the highest withdrawal resistance, followed by fasteners in the T direction. In this regard, obtaining data about various fasteners in all loading directions could be valuable for finding optimal fasteners for each direction.-Different layer arrangements were examined to improve the low withdrawal resistance in the L direction of the CLT. The results showed that the difference between withdrawal resistance in L and T directions was reduced by changing the arrangement, thereby improving them in the L direction.-Failure modes in different CLT directions and different fastener types should be considered for achieving a better withdrawal resistance. Further research is recommended to characterize the impact of thread height and gauge (flank distance).-The higher the diameter of fasteners, the higher the damaged area for each fastener, which directly correlates with withdrawal capacity. The damaged area in the S direction was higher than in the T and L directions. Therefore, applying these findings to the design of angle brackets connected with nails or screws or a combination of them is recommended.-The results revealed that screws with larger diameters showed high withdrawal resistance. However, it is important to consider where to install these fasteners on the CLT because of how much more damage they might do. The design of the angle brackets could benefit from these insights. Therefore, using fasteners with smaller diameters in the low end and edge distances of the angle brackets and fasteners with larger diameters in the higher end and edge distances is recommended.

## Figures and Tables

**Figure 1 polymers-14-03129-f001:**
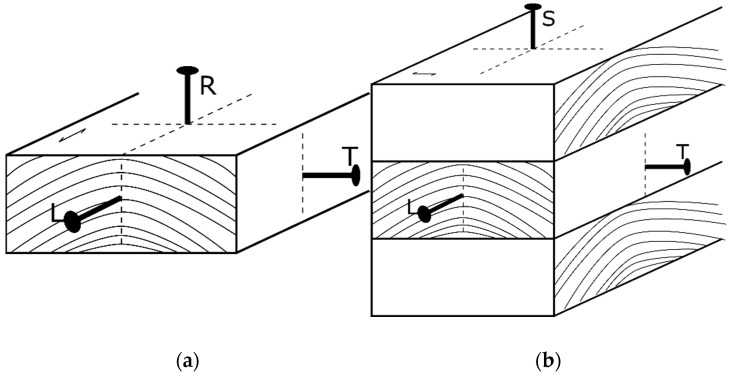
(**a**) Withdrawal directions in sawn timber: (L) parallel to the grain or longitudinal, (R) perpendicular to the grain in radial direction, (T) perpendicular to the grain in tangential direction (side face). (**b**) Loading direction in CLT: (L) longitudinal edge loading (parallel to grain), (T) tangential edge loading (perpendicular to grain), (S) perpendicular to the surface loading.

**Figure 2 polymers-14-03129-f002:**
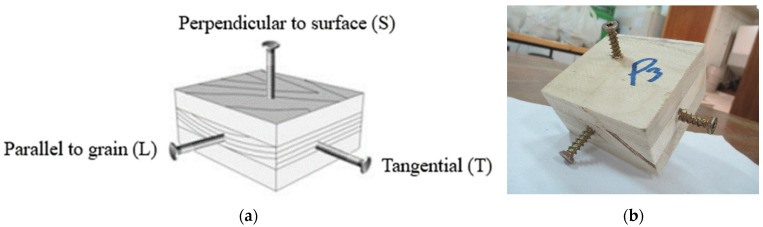
Loading directions (**a**) and placements of fasteners (**b**) in the CLT samples for the withdrawal tests.

**Figure 3 polymers-14-03129-f003:**
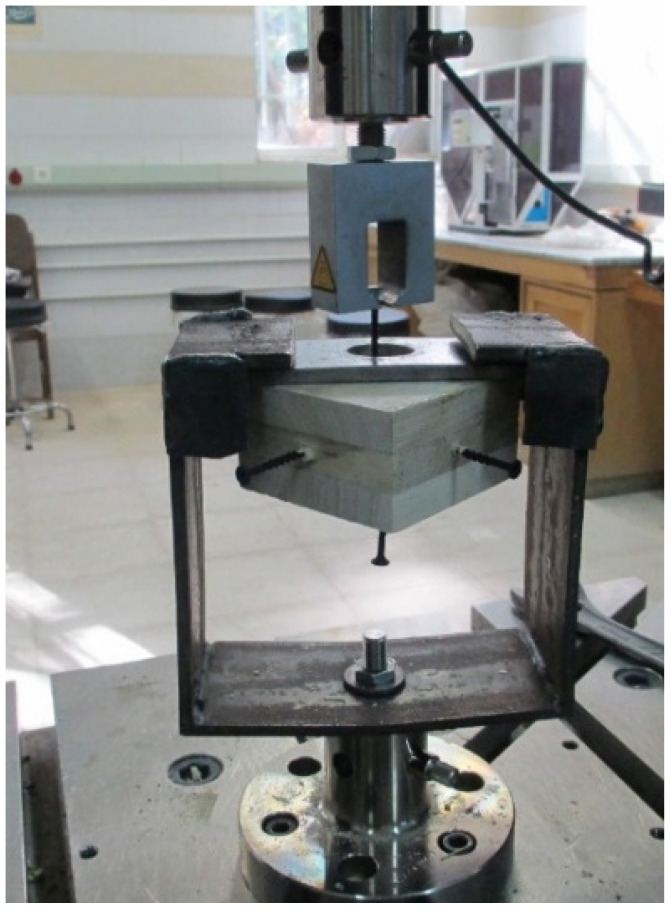
Instron machine for testing withdrawal resistance of the fasteners.

**Figure 4 polymers-14-03129-f004:**
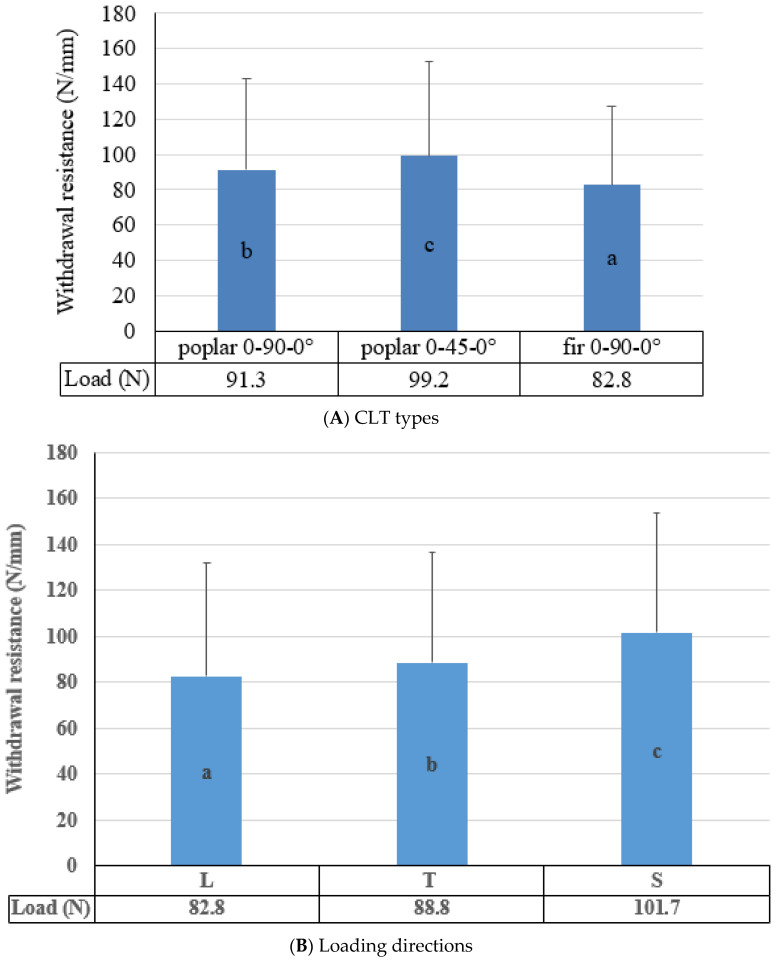
Effects of layer arrangements (**A**), loading directions (**B**), and types of fasteners (**C**) on withdrawal resistance.

**Figure 5 polymers-14-03129-f005:**
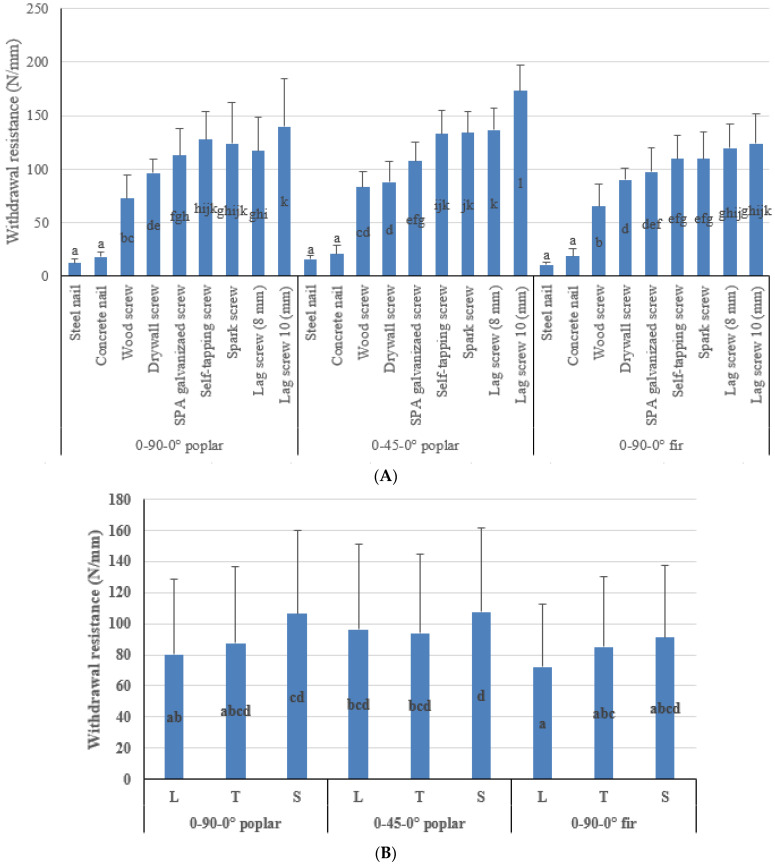
Interaction effects layer arrangements, loading directions, and types of fasteners on the withdrawal resistance of CLTs. (**A**) CLT types * Fastener types. (**B**) CLT types * Loading directions. (**C**) Fastener types * Loading directions.

**Figure 6 polymers-14-03129-f006:**
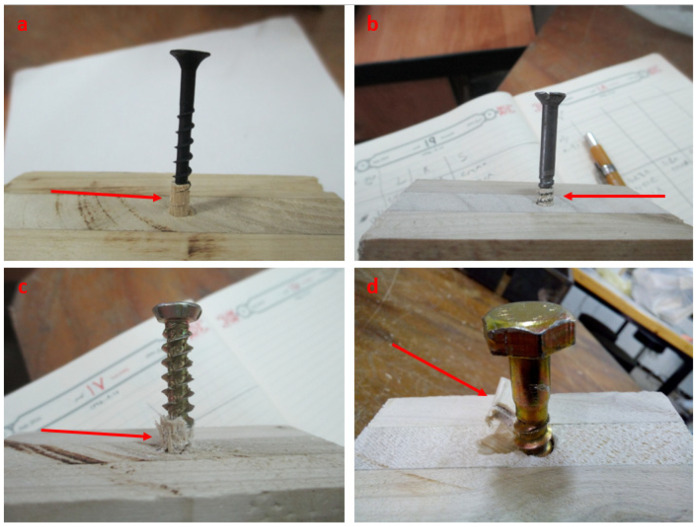
Typical failure modes of CLT members under the withdrawal load parallel to the grains (L direction). (**a**–**c**) Column-like pull-out of wood fibers, (**d**) inclined shear failure along with the grains.

**Figure 7 polymers-14-03129-f007:**
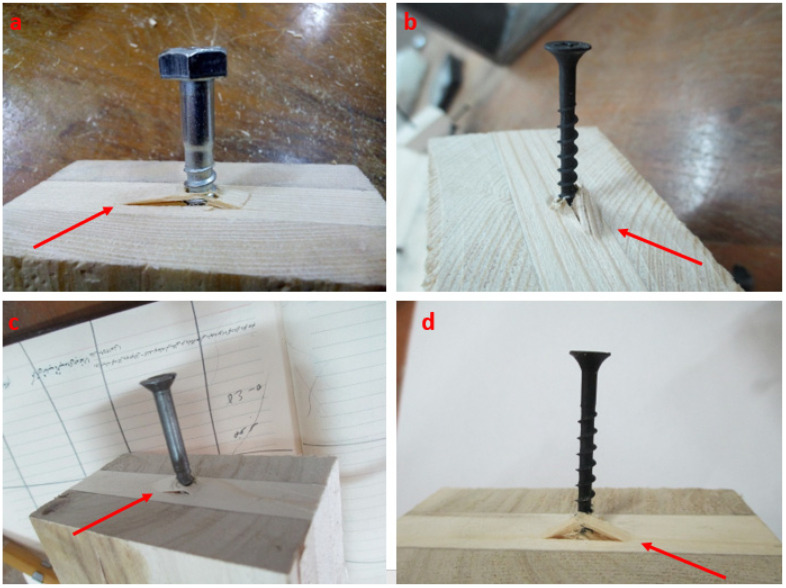
Typical failure modes of CLT members under the withdrawal load parallel to the grains (T direction). (**a**–**d**) Tearing of adjacent fibers around the fasteners.

**Figure 8 polymers-14-03129-f008:**
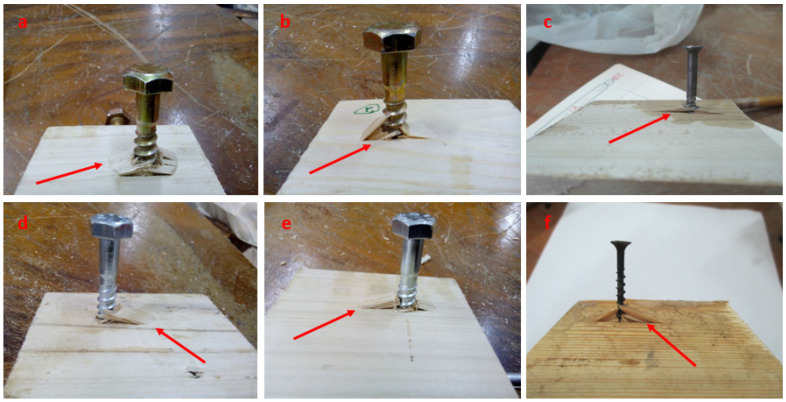
Typical failure modes of CLT members under the withdrawal load parallel to the grains (S direction). (**a**–**d**) Withdrawal failure accompanied by substantial fiber deformation.

**Table 1 polymers-14-03129-t001:** Characteristics of fasteners used for withdrawal tests in CLT members.

	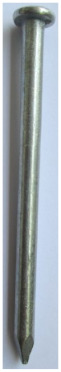	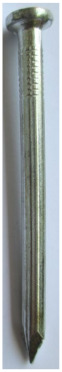	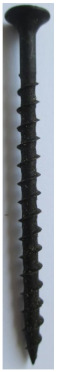	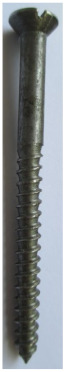	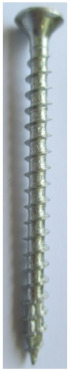	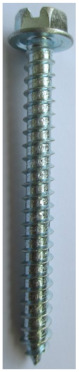	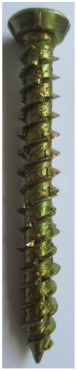	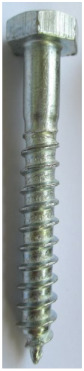	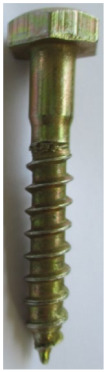
	Steel Nail	Concrete Nail	Drywall Screw	Wood Screw	Galvanized SPAX Screw	Self-Tapping Screw	Concrete Screw	Lag Screw(8 mm)	Lag Screw(10 mm)
**Length (mm)**	63.5	66	61.8	62.5	59.16	68.7	60	65.5	69
**Major diameter (mm)**	3.75	4.16	4.2	4.7	4.46	6.44	7.8	7.75	9.43
**Minor diameter (mm)**	-	-	2.7	3.9	2.9	4.7	4.7	5.6	6.6
**Pre-drilled hole diameter (mm)**	3	3	2	3.5	2	3.5	3.5	5	6.5

**Table 2 polymers-14-03129-t002:** Withdrawal resistance of fasteners in all arrangements and loading directions.

Fastener	Loading Direction	Withdrawal Resistance (N)		
		Poplar (0-90-0°)	Poplar (0-45-0°)	Fir (0-90-0°)
**Steel nail**	**L**	**10** (0.6)	**14** (2.7)	**9** (1.5)
	**T**	**14** (4)	**15** (1.6)	**13** (3.3)
	**S**	**13** (4.3)	**19** (3.5)	**11** (2.4)
**Concrete nail**	**L**	**18** (2.3)	**17** (3.3)	**21** (7.2)
	**T**	**14** (2.9)	**23** (8.9)	**16** (4.7)
	**S**	**23** (3.7)	**23** (10.2)	**19** (7.9)
**Wood screw**	**L**	**47** (9.2)	**77** (11.4)	**47** (12.6)
	**T**	**77** (5.9)	**79** (10.4)	**73** (23)
	**S**	**95** (10.2)	**93** (17.3)	**75** (14.3)
**Drywall screw**	**L**	**92** (15.9)	**82** (19.7)	**86** (16.5)
	**T**	**98** (16)	**79** (17.8)	**94** (8.6)
	**S**	**99** (7.2)	**104** (7.3)	**91** (5.5)
**SPAX galvanized screw**	**L**	**83** (10.6)	**98** (18.7)	**71** (6)
	**T**	**127** (18.5)	**108** (21.3)	**106** (15.6)
	**S**	**129** (10.4)	**116.1** (10.3)	**115** (10)
**Self-tapping screw**	**L**	**97** (14)	**123** (16.1)	**91** (13.6)
	**T**	**140** (5.8)	**123** (20.8)	**119** (22.9)
	**S**	**147** (19.9)	**153** (13.3)	**119** (18.2)
**Concrete** **screw**	**L**	**89** (42)	**146** (15.5)	**76** (6)
	**T**	**152** (25.7)	**118** (18.7)	**127** (8)
	**S**	**131** (11.7)	**138** (14.4)	**126** (6)
**Lag screw (8 mm)**	**L**	**126** (9.4)	**126** (14.4)	**137** (8)
	**T**	**78** (10.5)	**134** (20)	**98** (26)
	**S**	**147** (6.2)	**150** (20)	**124** (9.2)
**Lag screw (10 mm)**	**L**	**159** (10.1)	**182** (20.8)	**111** (19.3)
	**T**	**89** (32.5)	**161** (23.3)	**119** (37.7)
	**S**	**172** (24.7)	**172** (29.2)	**141** (18.5)

The numbers in parenthesis show standard deviation.

**Table 3 polymers-14-03129-t003:** Analysis of variance’s main and interaction effects of the layer arrangement, fastener type, loading direction on the withdrawal resistance of CLTs.

Source	Type III Sum of Squares	df	Mean Square	F	Sig.
CLT types	21,815.903	2	10,907.951	45.480	0.000 **
Fastener types	965,774.335	8	120,721.792	503.337	0.000 **
Loading directions	30,160.979	2	15,080.490	62.877	0.000 **
CLT types * Fastener types	22,264.690	16	1391.543	5.802	0.000 **
CLT types * Loading directions	5502.278	4	1375.569	5.735	0.000 **
Fastener types * Loading directions	41,220.657	16	2576.291	10.742	0.000 **
CLT types * Fastener types * Loading directions	43,430.328	32	1357.198	5.659	0.000 **

****** significant at 99% confident level; ***** significant at 95% confident level; ns: not significant.

## Data Availability

The data presented in this study are available on request from the corresponding author.

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
