# Peer review of "Withdrawal Performance of Nails and Screws in Cross-Laminated Timber (CLT) Made of Poplar (Populus alba) and Fir (Abies alba)"

_polymers, 2022, doi:10.3390/polym14153129_

Round 1
Reviewer 1 Report
The article offers an interesting way of researching the tensile performance of nails and screws in cross-laminated timber (CLT) from poplar and fir.
I ask the authors to supplement the abstract with the aim of the article.
The introduction chapter is interestingly written with the purpose of introducing the reason for the research. But again, the goal of the article should be stated and specified at the end of the introduction chapter.
It is necessary to specify the selection of samples. Why poplar and fir?
Why are the fig. 4a) and fig. 5 a), b) the given results of spruce? How does this relate to the research presented?
Author Response
|
Reviewer’s comment: I ask the authors to supplement the abstract with the aim of the article. Authors’ response: Thanks for your comment. The supplementary sentence has been added to the abstract part. |
|
|
|
Reviewer’s comment: The introduction chapter is interestingly written with the purpose of introducing the reason for the research. But again, the goal of the article should be stated and specified at the end of the introduction chapter. Authors’ response: Thanks for your comment. The correction has been made at the end of the introduction part. |
|
|
|
Reviewer’s comment: It is necessary to specify the selection of samples. Why poplar and fir? Authors’ response: Thanks for your comment. Poplar is a fast growing and hardwood species. Fir is one of the common and softwood species in manufacturing CLT. Both of the have similar density. In countries with shortage of wood resources, fast growing species could be useful. As a result, the findings of poplar were compared to data obtained from CLTs made of Fir, a common softwood species used in CLT. These details are provided in the introduction and materials and methods sections. |
|
|
|
Reviewer’s comment: Why are the fig. 4a) and fig. 5 a), b) the given results of spruce? How does this relate to the research presented?
Authors’ response: Thanks for your comment. All of them are fir. The correction has been made to the manuscript. |
Reviewer 2 Report
The presented manuscript is focused on investigating and evaluating the withdrawal resistance of nine types of conventional fasteners in the three main directions of three-layer cross laminated timber fabricated from poplar and fir wood. In general, the manuscript is well-written, structured and informative, and can be accepted for publication in the Polymers Journal after some revisions. Please check my comments on your work, presented below:
General remark: the manuscript has not been prepared according to the journal template and requirements. The pages and lines have not been numbered which makes the review process a bit difficult. Please check the Instructions for authors.
Overall, the title, abstract and the keywords correspond to the aims and objectives of the manuscript.
The abstract of the manuscript and the keywords correspond to the aims and objectives of the paper. The abstract is informative, and contains the main findings of the article. However, the addition of more concrete results of your research is recommended.
page 1: in the abstract, and onwards in the manuscript: there is no need to use capital letters for poplar and fir. The same comments also apply to other species mentioned in the manuscript (larch, spruce, radiata pine, etc.).
Please add also the botanical name of the species.
page 2: I’d recommend to delete the section “Research highlights”, it is not required.
page 2: once introduced the abbreviation EWP, please use it instead of the full term “engineered wood product”.
page 4: please define “CLT side face” and “CLT narrow face”.
page 5: what does “karamatsu” mean?
page 6: “areas with a shortage of wood supplies, such as Iran” – this statement might be true for Iran, but also for almost all of the world, nowadays the shortage of wood raw materials is a global problem.
page 6: “threads in fasteners make them simple to drive yet tough to disengage.” – please rephrase, the statement is not very clear as it is.
page 8, Materials and Methods: “2000 × 110 × 25 mm” should be “2000 mm × 110 mm × 25 mm”, please revise.
page 8: “boards without any faults” – I’d recommend to replace faults with wood defects, it is more appropriate term.
page 9: “75 × 75 × 48 mm” should be “75 mm × 75mm × 48 mm”, please revise.
page 8, fig. 2b is not of a very good quality, please replace it with a better picture, if possible.
page 9: please define “boundary conditions”.
page 11: please provide relevant information on the testing machine used (company producer, city, country).
page 15: where did you get the respective values of withdrawal resistance of 91.3, 99.2, and 82.8? I couldn’t find them in Table 2.
page 16: please revise the captions of fig. 4 by adding the respective titles of a), b), and c).
page 18: “the withdrawal resistance of steel nails, concrete nails, wood screws, lag screws (8 mm), and lag screws (10 mm) changed 27.2 %, 16.1 %, 13.9 %, 16.8 %, and 23.8 %, respectively” – it is not very meaningful or justified to compare nails and screws. It would be better to make separate comparisons, e.g. between the two types of nails and different types of screws tested.
The Conclusion part reflects the main findings of the manuscript. However, the first three sentences are not needed since this information has already been given in the abstract of your work, please delete/revise them.
The References cited are appropriate to the research topic, but are not formatted according to the Journal requirements. Please refer to the Instructions for Authors.
Best regards!
Author Response
|
Reviewer’s comment: General remark: the manuscript has not been prepared according to the journal template and requirements. The pages and lines have not been numbered which makes the review process a bit difficult. Please check the Instructions for authors. Authors’ response: Thanks for your comment. Our apologies for the inconvenience. Page and line number have been added to the manuscript. |
|
|
|
Reviewer’s comment: The abstract of the manuscript and the keywords correspond to the aims and objectives of the paper. The abstract is informative, and contains the main findings of the article. However, the addition of more concrete results of your research is recommended. Authors’ response: Thanks for your comment. More concrete and quantitative results have been added to the abstract section. |
|
|
|
Reviewer’s comment: page 1: in the abstract, and onwards in the manuscript: there is no need to use capital letters for poplar and fir. The same comments also apply to other species mentioned in the manuscript (larch, spruce, radiata pine, etc.) Authors’ response: Thanks for your comment. The correction has been made in the manuscript. |
|
|
|
Reviewer’s comment: Please add also the botanical name of the species. Authors’ response: Thanks for your comment. The correction has been made in the manuscript. |
|
|
|
Reviewer’s comment: page 2: I’d recommend to delete the section “Research highlights”, it is not required. Authors’ response: Thanks for your comment. This section has been removed from the manuscript. |
|
|
|
Reviewer’s comment: page 2: once introduced the abbreviation EWP, please use it instead of the full term “engineered wood product”. Authors’ response: Thanks for your comment. The correction has been made in the manuscript. |
|
|
|
Reviewer’s comment: page 4: please define “CLT side face” and “CLT narrow face”. Authors’ response: Thanks for your comment. Details have been added to the manuscript. |
|
|
|
Reviewer’s comment: page 5: what does “karamatsu” mean? Authors’ response: Thanks for your comment. The correction has been made in the manuscript. |
|
|
|
Reviewer’s comment: page 6: “areas with a shortage of wood supplies, such as Iran” – this statement might be true for Iran, but also for almost all of the world, nowadays the shortage of wood raw materials is a global problem. Authors’ response: Thanks for your comment. Details have been added to the manuscript. |
|
|
|
Reviewer’s comment: page 6: “threads in fasteners make them simple to drive yet tough to disengage.” – please rephrase, the statement is not very clear as it is. Authors’ response: Thanks for your comment. The correction has been made in the manuscript. |
|
|
|
Reviewer’s comment: page 8, Materials and Methods: “2000 × 110 × 25 mm” should be “2000 mm × 110 mm × 25 mm”, please revise. Authors’ response: Thanks for your comment. The correction has been made in the manuscript. |
|
|
|
Reviewer’s comment: page 8: “boards without any faults” – I’d recommend to replace faults with wood defects, it is more appropriate term. Authors’ response: Thanks for your comment. The correction has been made in the manuscript. |
|
|
|
Reviewer’s comment: page 9: “75 × 75 × 48 mm” should be “75 mm × 75mm × 48 mm”, please revise. Authors’ response: Thanks for your comment. The correction has been made in the manuscript. |
|
|
|
Reviewer’s comment: page 8, fig. 2b is not of a very good quality, please replace it with a better picture, if possible. Authors’ response: Thanks for your comment. The replacement has been done in the manuscript. |
|
|
|
Reviewer’s comment: page 9: please define “boundary conditions”. Authors’ response: Thanks for your comment. In this research, it means that the distance between the screws in different directions is such that they do not affect each other. Details have been added to the manuscript. |
|
|
|
Reviewer’s comment: page 11: please provide relevant information on the testing machine used (company producer, city, country). Authors’ response: Thanks for your comment. Details have been added to the manuscript. |
|
|
|
Reviewer’s comment: page 15: where did you get the respective values of withdrawal resistance of 91.3, 99.2, and 82.8? I couldn’t find them in Table 2. Authors’ response: Thanks for your comment. Table 2 is just the withdrawal resistance of the fasteners regardless of the statistical analysis (The main and interaction effects). 91.3, 99.2, and 82.8, are the main effects or interaction effects of the variables. · A main effect is the effect of a single independent variable on a dependent variable – ignoring all other independent variables. · An interaction effect is the simultaneous effect of two or more independent variables on at least one dependent variable in which their joint effect is significantly greater (or significantly less) than the sum of the parts. As a result, the numbers are not the same. |
|
|
|
|
|
|
|
Reviewer’s comment: page 16: please revise the captions of fig. 4 by adding the respective titles of a), b), and c). Authors’ response: Thanks for your comment. The correction has been made in the manuscript. |
|
|
|
Reviewer’s comment: page 18: “the withdrawal resistance of steel nails, concrete nails, wood screws, lag screws (8 mm), and lag screws (10 mm) changed 27.2 %, 16.1 %, 13.9 %, 16.8 %, and 23.8 %, respectively” – it is not very meaningful or justified to compare nails and screws. It would be better to make separate comparisons, e.g. between the two types of nails and different types of screws tested. Authors’ response: Thanks for your comment. The changes have been made in the manuscript according to the comment.
|
|
|
|
Reviewer’s comment: The Conclusion part reflects the main findings of the manuscript. However, the first three sentences are not needed since this information has already been given in the abstract of your work, please delete/revise them. Authors’ response: Thanks for your comment. This section has been deleted from the manuscript. |
|
|
|
Reviewer’s comment: The References cited are appropriate to the research topic, but are not formatted according to the Journal requirements. Please refer to the Instructions for Authors. Authors’ response: Thanks for your comment. The correction has been made in the manuscript according to the journal style. |
Reviewer 3 Report
Just reporting about results, but no scientific discission.

Author Response
|
Reviewer’s comment: There is no indication of pages or lines Authors’ response: Thanks for your comment. The correction has been made in the manuscript. |
||||||||||||||||||||||||||||||||||||||||||||||||||||||||||||||||||||||||||||||||||||||||||||||||
|
|
||||||||||||||||||||||||||||||||||||||||||||||||||||||||||||||||||||||||||||||||||||||||||||||||
|
Reviewer’s comment: poplar, fir, larch, spruce, always with small letters at the beginning of the word Authors’ response: Thanks for your comment. The correction has been made in the manuscript. |
||||||||||||||||||||||||||||||||||||||||||||||||||||||||||||||||||||||||||||||||||||||||||||||||
|
|
||||||||||||||||||||||||||||||||||||||||||||||||||||||||||||||||||||||||||||||||||||||||||||||||
|
Reviewer’s comment: What is “CLT side face” and “CLT narrow face”? Authors’ response: Thanks for your comment. Details have been added to the manuscript. |
||||||||||||||||||||||||||||||||||||||||||||||||||||||||||||||||||||||||||||||||||||||||||||||||
|
|
||||||||||||||||||||||||||||||||||||||||||||||||||||||||||||||||||||||||||||||||||||||||||||||||
|
Reviewer’s comment: What means “column-like pull out of timber components”? Authors’ response: Thanks for your comment. This expression is according to this paper: Withdrawal resistance of self-tapping screws inserted on the narrow face of cross laminated timber made from Radiata Pine |
||||||||||||||||||||||||||||||||||||||||||||||||||||||||||||||||||||||||||||||||||||||||||||||||
|
|
||||||||||||||||||||||||||||||||||||||||||||||||||||||||||||||||||||||||||||||||||||||||||||||||
|
Reviewer’s comment: What means “threads in fasteners make them simple to drive yet tough to disengage”? Authors’ response: Thanks for your comment. It means threads make fasteners easy to insert and hard to pull-out .The correction has been made in the manuscript. |
||||||||||||||||||||||||||||||||||||||||||||||||||||||||||||||||||||||||||||||||||||||||||||||||
|
|
||||||||||||||||||||||||||||||||||||||||||||||||||||||||||||||||||||||||||||||||||||||||||||||||
|
Reviewer’s comment: Use SI units, e.g., kg/m³ instead of gr/cm³. Authors’ response: Thanks for your comment. The correction has been made in the manuscript accordingly. |
||||||||||||||||||||||||||||||||||||||||||||||||||||||||||||||||||||||||||||||||||||||||||||||||
|
|
||||||||||||||||||||||||||||||||||||||||||||||||||||||||||||||||||||||||||||||||||||||||||||||||
|
Reviewer’s comment: Either e.g., 90/0/90 or 90-0-90, but not alternatively using Authors’ response: Thanks for your comment. These are according to the arrangements layers for manufacturing the CLTs. In other words, in these arrangements the layers placed on each other in 90°. For 0-45-0, the middle layer is in 45° to the upper and bottom layer. There is no 90-0-90 arrangement.
|
||||||||||||||||||||||||||||||||||||||||||||||||||||||||||||||||||||||||||||||||||||||||||||||||
|
|
||||||||||||||||||||||||||||||||||||||||||||||||||||||||||||||||||||||||||||||||||||||||||||||||
|
Reviewer’s comment: What means: “The fasteners were placed to eliminate any gaps and to meet the requirements of boundary conditions“? Authors’ response: Thanks for your comment. In this research, it means that the distance between the screws in different directions is such that they do not affect each other. Details have been added to the manuscript. |
||||||||||||||||||||||||||||||||||||||||||||||||||||||||||||||||||||||||||||||||||||||||||||||||
|
|
||||||||||||||||||||||||||||||||||||||||||||||||||||||||||||||||||||||||||||||||||||||||||||||||
|
Reviewer’s comment: How was the direction of the fastener in case of 0-45-0 CLT? Authors’ response: Thanks for your comment. All fasteners inserted vertically on each direction (0-90-0°, and 0-45-0° means that the middle layer of CLTs in 45° to the upper and bottom layers. |
||||||||||||||||||||||||||||||||||||||||||||||||||||||||||||||||||||||||||||||||||||||||||||||||
|
|
||||||||||||||||||||||||||||||||||||||||||||||||||||||||||||||||||||||||||||||||||||||||||||||||
|
Reviewer’s comment: “Fir CLTs with one-layer arrangement (0-90-0°)”: => “with only one arrangement of layers…” Authors’ response: Thanks for your comment. The correction has been made in the manuscript. |
||||||||||||||||||||||||||||||||||||||||||||||||||||||||||||||||||||||||||||||||||||||||||||||||
|
|
||||||||||||||||||||||||||||||||||||||||||||||||||||||||||||||||||||||||||||||||||||||||||||||||
|
Reviewer’s comment: “Fasteners in Poplar CLTs with an 0-45-0° arrangement had the maximum withdrawal resistance”. In which direction of insertion of the fastener? Authors’ response: Thanks for your comment. According to table 2, generally, in all loading direction (L, T, and S) in the arrangement of 0-45-0, fasteners showed more withdrawal resistance compared to other direction. Figure 1 shows the schematic of fastener in CLTs. |
||||||||||||||||||||||||||||||||||||||||||||||||||||||||||||||||||||||||||||||||||||||||||||||||
|
|
||||||||||||||||||||||||||||||||||||||||||||||||||||||||||||||||||||||||||||||||||||||||||||||||
|
Reviewer’s comment: Rethink the number of significant digits; e.g., 126.6 at a standard deviation of 18.53 are too many digits. Authors’ response: Thanks for your comment. The correction has been made accordingly. |
||||||||||||||||||||||||||||||||||||||||||||||||||||||||||||||||||||||||||||||||||||||||||||||||
|
|
||||||||||||||||||||||||||||||||||||||||||||||||||||||||||||||||||||||||||||||||||||||||||||||||
|
“According to Fig. 4a, Poplar CLTs with the 0-45-0° arrangement”: in which direction? Authors’ response: Thanks for your comment. By considering factors, including all fastener type and all loading direction, this result obtained from the SPSS software statistical analysis. Figure 4 (a, b, and c) show the main effects. · A main effect is the effect of a single independent variable on a dependent variable – ignoring all other independent variables.
|
||||||||||||||||||||||||||||||||||||||||||||||||||||||||||||||||||||||||||||||||||||||||||||||||
|
|
||||||||||||||||||||||||||||||||||||||||||||||||||||||||||||||||||||||||||||||||||||||||||||||||
|
Reviewer’s comment: “According to Fig. 4a, Poplar CLTs with the 0-45-0° arrangement had the highest withdrawal resistance (99.2 N)“: from where is this result 99.2 N? It is not mentioned in Table 2. Authors’ response: Thanks for your comment. Table 2 is just the withdrawal resistance of the fasteners regardless of the statistical analysis (The main and interaction effects). 91.3, 99.2, and 82.8, are the main effects or interaction effects of the variables. · A main effect is the effect of a single independent variable on a dependent variable – ignoring all other independent variables. · An interaction effect is the simultaneous effect of two or more independent variables on at least one dependent variable in which their joint effect is significantly greater (or significantly less) than the sum of the parts. As a result, the numbers are not the same. |
||||||||||||||||||||||||||||||||||||||||||||||||||||||||||||||||||||||||||||||||||||||||||||||||
|
|
||||||||||||||||||||||||||||||||||||||||||||||||||||||||||||||||||||||||||||||||||||||||||||||||
|
Reviewer’s comment: What means the thin black line above the blue bar? The length of this line is partly two thirds of the blue bar. Is it a standard deviation? But for which result? This is the same in many Figures. Authors’ response: Thanks for your comment. Yes, they are standard deviation. Here is output of statistical analysis for figure 4 (a, b and c).
|
||||||||||||||||||||||||||||||||||||||||||||||||||||||||||||||||||||||||||||||||||||||||||||||||
|
|
||||||||||||||||||||||||||||||||||||||||||||||||||||||||||||||||||||||||||||||||||||||||||||||||
|
Reviewer’s comment: “The presence of parenchyma rays in hardwoods like Poplar may explain this difference [45], and it is hypothesized that these rays resist the withdrawal of fasteners.“ This should be explained more in detail, because this is the scientific part of the paper.
Authors’ response: Thanks for your comment. More details have been added to the manuscript. |
||||||||||||||||||||||||||||||||||||||||||||||||||||||||||||||||||||||||||||||||||||||||||||||||
|
|
||||||||||||||||||||||||||||||||||||||||||||||||||||||||||||||||||||||||||||||||||||||||||||||||
|
Reviewer’s comment: Is there really a statistically significant difference between L and T? This seems to be the case only for selected fasteners, not for all.
Authors’ response: Thanks for your comment. Yes, according to figure 4 (b) there is a significant difference between them. |
||||||||||||||||||||||||||||||||||||||||||||||||||||||||||||||||||||||||||||||||||||||||||||||||
|
|
||||||||||||||||||||||||||||||||||||||||||||||||||||||||||||||||||||||||||||||||||||||||||||||||
|
Reviewer’s comment: From where are the results shown in this Figure?
Authors’ response: Thanks for your comment. All results of figure 4 (a, b and c) are the main effects of variables. As a results, they will be different from the table 2. |
||||||||||||||||||||||||||||||||||||||||||||||||||||||||||||||||||||||||||||||||||||||||||||||||
|
|
||||||||||||||||||||||||||||||||||||||||||||||||||||||||||||||||||||||||||||||||||||||||||||||||
|
Reviewer’s comment: Data for which CLT type, which direction? This must be indicated.
Authors’ response: Thanks for your comment. By considering all CLT types, these results obtained from the SPSS statistical software. |
||||||||||||||||||||||||||||||||||||||||||||||||||||||||||||||||||||||||||||||||||||||||||||||||
|
|
||||||||||||||||||||||||||||||||||||||||||||||||||||||||||||||||||||||||||||||||||||||||||||||||
|
Reviewer’s comment: “Concrete nails feature a low-profile thread on their shanks, providing higher withdrawal resistance than steel nails, which do not have any threads.” Comparing nails and screws give no sense. Please compare only the various screws among them (or only the nails among them) in terms of shape/size/surface/threads.
Authors’ response: Thanks for your comment. Some details have been added to the manuscript accordingly. |
||||||||||||||||||||||||||||||||||||||||||||||||||||||||||||||||||||||||||||||||||||||||||||||||
|
|
||||||||||||||||||||||||||||||||||||||||||||||||||||||||||||||||||||||||||||||||||||||||||||||||
|
Reviewer’s comment: What is the “flank distance”?
Authors’ response: Thanks for your comment. The flank distance is the axial distance between a thread flank and the immediately adjacent flank. |
||||||||||||||||||||||||||||||||||||||||||||||||||||||||||||||||||||||||||||||||||||||||||||||||
|
|
||||||||||||||||||||||||||||||||||||||||||||||||||||||||||||||||||||||||||||||||||||||||||||||||
|
Reviewer’s comment: For such comparisons (such as “27.2 %, 16.1 %, 13.9 %, 16.8 %, and 23.8 %”) use only full digit numbers.
Authors’ response: Thanks for your comment. The correction has been made in the manuscript. |
||||||||||||||||||||||||||||||||||||||||||||||||||||||||||||||||||||||||||||||||||||||||||||||||
|
|
||||||||||||||||||||||||||||||||||||||||||||||||||||||||||||||||||||||||||||||||||||||||||||||||
|
Reviewer’s comment: “CLTs with the 0-45-0°-layer arrangement in contact with the fasteners was the same (45°) under both loading directions (L and T);”: this should be shown in an additional figure, because that is not clear.
Authors’ response: Thanks for your comment. Figure 6 (d) shows the failure mode of that. In order to make more clarity, some details have been added to that sentence. The below schematic might make more clarity. |
||||||||||||||||||||||||||||||||||||||||||||||||||||||||||||||||||||||||||||||||||||||||||||||||
|
|
||||||||||||||||||||||||||||||||||||||||||||||||||||||||||||||||||||||||||||||||||||||||||||||||
|
Reviewer’s comment: “Conversely, for hardwoods such as Poplar, the significant difference between S and T directions may be related to the placement of the parenchyma rays in the radial direction of the wood and perpendicular to the annual rings.” But rays are also given in softwood such as fir.
Authors’ response: Thanks for your comment. rays are also given in softwood such as fir. However, they are single row. Multiple row rays are more common among hardwoods. Generally, poplar has rays with more than single row (may be two rows). Moreover, width of the rays in hardwoods are more than in softwoods. Those results might be related to this.
|
||||||||||||||||||||||||||||||||||||||||||||||||||||||||||||||||||||||||||||||||||||||||||||||||
|
|
||||||||||||||||||||||||||||||||||||||||||||||||||||||||||||||||||||||||||||||||||||||||||||||||
|
Reviewer’s comment: Either shorten text in this chapter of eliminate Fig. 5; it is simply too long.
Author’s response: Thanks for your comment. This section contains three diagrams, indicating the interaction effects of all variables. Therefore, the text is long. The authors ask you kindly to let keep all of them, since they convey valuable data in terms of all factors. The text contains many details in terms of all variables. |
||||||||||||||||||||||||||||||||||||||||||||||||||||||||||||||||||||||||||||||||||||||||||||||||
|
|
||||||||||||||||||||||||||||||||||||||||||||||||||||||||||||||||||||||||||||||||||||||||||||||||
|
Reviewer’s comment: In other words, in this direction, the areas of the failures (Fig 7a, b, c, and d) were larger than those in the L direction (Fig 6a, b, c, and d). This sentence is correct, but it is at the wrong place. In this paragraph you talk about S-direction.
Authors’ response: Thanks for your comment. This sentence is unclear. The correction has been made in the manuscript. in this direction (S), the areas of the failures (Fig 7a, b, c, and d) were larger than those in the L direction (Fig 6a, b, c, and d). |
||||||||||||||||||||||||||||||||||||||||||||||||||||||||||||||||||||||||||||||||||||||||||||||||
|
|
||||||||||||||||||||||||||||||||||||||||||||||||||||||||||||||||||||||||||||||||||||||||||||||||
|
Reviewer’s comment: Further research to fully characterize the impact of thread height and thread gauge (flank distance) is recommended. But you have all data available. Why has this not been evaluated immediately?
Authors’ response: Thanks for your comment. The effects of thread height and thread gauge (flank distance)will be investigated through finite element modelling (FEM) in another research paper. |
||||||||||||||||||||||||||||||||||||||||||||||||||||||||||||||||||||||||||||||||||||||||||||||||
|
|
||||||||||||||||||||||||||||||||||||||||||||||||||||||||||||||||||||||||||||||||||||||||||||||||
|
Reviewer’s comment: Changing the direction of the fastener might have a positive consequence for withdrawal force. But usually the direction is determined by the application of the CLT, not the other way round. Please comment.
Authors’ response: Thanks for your comment. By having data about various fasteners in all direction, engineers can find the optimal way for using appropriate fasteners in terms of CLT direction. No need to use single type of fasteners in all directions. |
||||||||||||||||||||||||||||||||||||||||||||||||||||||||||||||||||||||||||||||||||||||||||||||||

Reviewer 4 Report
Withdrawal performance of nails and screws in cross-laminated timber (CLT) made of Poplar and Fir
In this study, the withdrawal resistance of nine types of conventional fasteners (stainless-steel nails, concrete nails and screws, drywall screws, three types of partially and fully threaded wood screws, and two types of lag screws), with three loading directions (parallel to the grain, perpendicular to the surface, and tangential), and two layer arrangements (0-90-0° and 0-45-0°) in 3-ply CLTs made of Poplar and Fir was investigated.
The article is well written and clear.
1. Abstract need revision with some quantitative results.
2. Some more latest studies are required in the introduction section to further highlight the importance of this study.
3. Section 2.1, what was the reason for the selected woods, why not other types?
4. Figure 2a, what is the Tangential, what is reference, grains or what?
5. What were the reasons for these fasteneres, any previous study with reference to these screwes.
6. Figure 3 could be revised with proper labels and appropriate size.
7. According to Fig. 4a, Poplar CLTs with the 0-45-0° arrangement had the highest withdrawal resistance (99.2 N), while those made from Fir wood had the lowest (82.8 N). What were the reasons?
8. Figure 4, what are a, b and c...., authors do not provide details before.
9. There is need to discuss results with scientific reasons especially failure modes.
Author Response
|
Reviewer #4
|
|
Reviewer’s comment: 1. Abstract need revision with some quantitative results.
Authors’ response: Thanks for your comment. Some quantitative results have been added to the abstract section. |
|
|
|
Reviewer’s comment: 2. Some more latest studies are required in the introduction section to further highlight the importance of this study.
Authors’ response: Thanks for your comment. Some related studies have been added to the manuscript. |
|
|
|
Reviewer’s comment: 3. Section 2.1, what was the reason for the selected woods, why not other types?
Author’s response: Thanks for your comment. Poplar is a fast growing and hardwood species. Fir is one of the common and softwood species in manufacturing CLT. Both of the have similar density. In countries with shortage of wood resources, fast growing species could be useful. As a result, the findings of poplar were compared to data obtained from CLTs made of Fir, a common softwood species used in CLT. These details are provided in the introduction and materials and methods sections. |
|
|
|
Figure 2a, what is the Tangential, what is reference, grains or what?
Author’s response: Thanks for your comment. It is a side face (surface) of CLTs. The correction has been made in the manuscript. |
|
|
|
Reviewer’s comment: 5. What were the reasons for these fasteners, any previous study with reference to these screws. Author’s response: Thanks for your comment. Most of the previous studies have been focused of self-tapping screws and nail. Some ones investigated the lag screws. But, in this research the focus was on these fasteners and other ones which have not been investigated. For example, concrete screw and nail screw can be use in this topic. Sometimes, CLTs will be installed on concrete, or these fasteners might be useful in timber-concrete composites. Therefore, this study provides new details regarding mentioned topics. |
|
|
|
Reviewer’s comment: 6. Figure 3 could be revised with proper labels and appropriate size.
Author’s response: Thanks for your comment. The correction has been made in the manuscript. |
|
|
|
Reviewer’s comment: 7. According to Fig. 4a, Poplar CLTs with the 0-45-0° arrangement had the highest withdrawal resistance (99.2 N), while those made from Fir wood had the lowest (82.8 N). What were the reasons?
Author’s response: Thanks for your comment. It might be due to the more involvement of fasteners with timber fibers in 0-45-0°. The details have been added to the manuscript. |
|
|
|
Reviewer’s comment: 8. Figure 4, what are a, b and c...., authors do not provide details before.
Author’s response: Thanks for your comment. All diagrams (4 and 5) provided the details of the results of the table 3. |
|
|
|
Reviewer’s comment: 9. There is need to discuss results with scientific reasons especially failure modes.
Author’s response: Thanks for your comment. In terms of discussion, especially failure modes, there is a lack of previous studies. However, We tries to discuss as much as we can according to the previous studies.
|
|
|

Round 2
Reviewer 3 Report
The authors made many changes and improvements according to the recommendations of the 4 reviewers.
However the main aspect remains: this paper is a test report, but no scientific paper. There was no addition of further scientific evaluation of the results. Still nails and screws are conmpared, despite the recommendation of all reviewers.
Ther must be a mistake in the statistical evaluation. The authors mentioned that most of results are significantly different. But looking at Figure 4 (a - c) with the huge standard deviations it is clear as immediately the appearance shows that there is no statistical difference in most cases, only between nails and screws. Standard deviations as shown in Fig. 4 are in the range of 50% and above; with such standard deviations you never get statistically secured difference, if the difference in the mean values is only 10 or 20%. Please check.
Author Response
Manuscript Number: polymers-1818057
Paper title: Withdrawal performance of nails and screws in cross-laminated timber (CLT) made of Poplar and Fir
Authors: Farshid Abdoli, Maria Rashidi, Akbar Rostampour-Haftkhani, Mohammad Layeghi, Ghanbar Ebrahimi
The authors thank the editor and the reviewers for giving us the opportunity to submit a revised version of our manuscript. We are also grateful to the editor and reviewers for the valuable time spent reviewing the paper and for invaluable comments. We have carefully addressed all comments. Here are the point-by-point responses to reviewers’ comments (Figures and details could be checked in attachment).
|
Reviewer #3
|
|
Regarding materials and methods, some details have been added to the manuscript. |
|
|
|
In order to support the results, the conclusion section has been rewritten. |
|
|
|
Reviewer’s comment: However, the main aspect remains: this paper is a test report, but no scientific paper. There was no addition of further scientific evaluation of the results. Still nails and screws are compared, despite the recommendation of all reviewers. Authors’ response: Thanks for your comment. Some details have been added to the manuscript. |
|
|
|
Reviewer’s comment: There must be a mistake in the statistical evaluation. The authors mentioned that most of results are significantly different. But looking at Figure 4 (a - c) with the huge standard deviations it is clear as immediately the appearance shows that there is no statistical difference in most cases, only between nails and screws. Standard deviations as shown in Fig. 4 are in the range of 50% and above; with such standard deviations you never get statistically secured difference, if the difference in the mean values is only 10 or 20%. Please check.
Authors’ response: Thanks for your comment. The difference in the withdrawal resistance between nails and screws was more than 11 times. When examining the main effects, the effect of the fastener type, which was the most influential factor, has been neglected in figures 4(a and b). As a result, the standard deviation is greatly increased, evident in figures 4(a and b). But the figure 4 (c), in which the main effect of the fastener type is investigated, the standard deviation of the bars is less. In addition, according to figure 5 (b), the fastener type effect has been neglected, and the standard deviation of the bars has increased. Hence, according to the mentioned details, the standard deviation of some figures is high.
According to the below paper, wood as an orthotropic material has significant anatomical variation in each direction (R, L, and T).
https://link.springer.com/article/10.1007/s00107-008-0294-9
It means that wood has various physical and mechanical properties in each direction. As a result, it is reasonable that the withdrawal resistance of fasteners in each direction has a significant difference.
Furthermore, in the current research, each direction showed various failure modes under the withdrawal load of fasteners in the failure modes section. In this regard, the (L) direction had the smallest failure mode area (the lowest withdrawal resistance), while the (R) direction had the largest one (the highest withdrawal resistance). The smallest failure mode area means less involvement between the fastener and timber part, resulting in the low withdrawal resistance of the fastener in that direction (L). On the other hand, the largest failure mode area means high involvement between the fastener and timber part, resulting in the high withdrawal resistance of the fastener in that direction (R). Therefore, the significant difference in the withdrawal resistance between all directions (R, L, and T) is reasonable.
Other studies also reported the same results. For instance:
https://www.sciencedirect.com/science/article/abs/pii/S2352012421001533
In this paper, the withdrawal resistance of fasteners has been evaluated in the side faces of CLT (L and T). different failure modes and withdrawal resistance have been reported in each loading direction.
Apparently, yes, it might seem that there is no significant difference between the bars in Figure 4 (a and b). But, statistical analysis shows that there is a significant difference between them. Here are the outputs according to the statistical analysis software:
Analysis of variance main and interaction effects of the layer arrangement, fastener type, loading direction on the withdrawal resistance of CLTs are as below:
Additionally, according to the tables below, Duncan’s test indicates a significant difference between groups (a, b, and c) (the left side table). Also, the standard deviation of the data is indicated in the right side table.
In terms of loading direction, Duncan’s test (the left side table) shows a significant difference between the groups (a, b, and c). Also, the standard deviation of the data is indicated in the right side table.
Regarding fastener type, Duncan’s test shows a significant difference between the groups (a, b, c, d, e, and g). However, no significant difference was observed for group f (self-tapping screw, concrete screw, and lag (8 mm)). Also, the standard deviation of the data is indicated in the below table:
And here are the interaction effects of the variables:
CLT type and fastener type:
CLT type and loading direction:
Fastener type and loading direction:
|

Round 3
Reviewer 3 Report
The authors continue to put nails and screws into one statistical population. With this all evaluations are distort and the results have no importance.
Author Response
Manuscript Number: polymers-1818057
Paper title: Withdrawal performance of nails and screws in cross-laminated timber (CLT) made of Poplar and Fir
Authors: Farshid Abdoli, Maria Rashidi, Akbar Rostampour-Haftkhani, Mohammad Layeghi, Ghanbar Ebrahimi
The authors thank the editor and the reviewers for giving us the opportunity to submit a revised version of our manuscript.
This study's main purpose was to evaluate the withdrawal resistance of various fasteners in all loading directions in poplar wood as a fast-growing species. There is a shortage of wooden resources. Hence, using fast-growing species is inevitable. Then, the results were compared to those in fir wood, a common species in manufacturing CLT. In this regard, comprehensive experimental tests (486 tests) were carried out. The results were also analyzed through a complete statistical analysis. Finally, the results indicated that poplar wood could be a common wood species for manufacturing CLT in terms of withdrawal resistance. We are working on these data in the next study through analytical approaches and finite element modelling.
